# Co-Delivery of a High Dose of Amphotericin B and Itraconazole by Means of a Dry Powder Inhaler Formulation for the Treatment of Severe Fungal Pulmonary Infections

**DOI:** 10.3390/pharmaceutics15112601

**Published:** 2023-11-08

**Authors:** Salomé S. Celi, Raquel Fernández-García, Andreina I. Afonso-Urich, M. Paloma Ballesteros, Anne Marie Healy, Dolores R. Serrano

**Affiliations:** 1Departamento de Farmacia Galénica y Tecnología Alimentaria, Facultad de Farmacia, Universidad Complutense de Madrid, Plaza Ramón y Cajal s/n, 28040 Madrid, Spain; 2Facultad de Farmacia, Instituto Universitario de Farmacia Industrial, Universidad Complutense de Madrid, Plaza de Ramón y Cajal s/n, 28040 Madrid, Spain; 3School of Pharmacy and Pharmaceutical Sciences, Trinity College Dublin, D02 PN40 Dublin, Ireland

**Keywords:** amphotericin B, itraconazole, dry powder inhaler, lung deposition, infection, aspergillosis

## Abstract

Over the past few decades, there has been a considerable rise in the incidence and prevalence of pulmonary fungal infections, creating a global health problem due to a lack of antifungal therapies specifically designed for pulmonary administration. Amphotericin B (AmB) and itraconazole (ITR) are two antifungal drugs with different mechanisms of action that have been widely employed in antimycotic therapy. In this work, microparticles containing a high dose of AmB and ITR (20, 30, and 40% total antifungal drug loading) were engineered for use in dry powder inhalers (DPIs) with an aim to improve the pharmacological effect, thereby enhancing the existing off-label choices for pulmonary administration. A Design of Experiment (DoE) approach was employed to prepare DPI formulations consisting of AmB-ITR encapsulated within γ-cyclodextrin (γ-CD) alongside functional excipients, such as mannitol and leucine. In vitro deposition indicated a favourable lung deposition pattern characterised by an upper ITR distribution (mass median aerodynamic diameter (MMAD) ~ 6 µm) along with a lower AmB deposition (MMAD ~ 3 µm). This offers significant advantages for treating fungal infections, not only in the lung parenchyma but also in the upper respiratory tract, considering that *Aspergillus* spp. can cause upper and lower airway disorders. The in vitro deposition profile of ITR and larger MMAD was related to the higher unencapsulated crystalline fraction of the drug, which may be altered using a higher concentration of γ-CD.

## 1. Introduction

Over the past decades, there has been a gradual rise in the incidence and prevalence of pulmonary fungal infections, primarily attributed to *Aspergillus* spp. [1,2,3]. The infection caused by *Aspergillus* spp. is usually asymptomatic; however, it can cause severe complications in immunocompromised individuals, including transplanted and oncology patients, and also in patients infected by the human immunodeficiency virus (HIV) [1,3,4]. These complications may also lead to high rates of mortality in immunodeficient individuals [2,5,6].

Amphotericin B (AmB) has been the gold-standard drug in antifungal therapy for more than 60 years [7,8,9]. This molecule selectively attaches to the ergosterol within the fungal cell membrane, inducing the formation of pores that enable the leakage of ions, primarily K^+^, which ultimately trigger apoptosis [7,10]. Today, AmBisome^®^ (Gilead Sciences Inc, Foster City, CA, USA), the liposomal formulation of AmB, is the most utilised AmB lipid-based formulation in clinical practice; however, all marketed AmB presentations are only designed and approved to be administered intravenously [1]. In 2006, the European Commission awarded orphan designation (EU/3/06/391) to Nektar Therapeutics UK Ltd. (London, UK), permitting the use of inhaled AmB to prevent pulmonary fungal infections in high-risk patients [1]. Currently, lipid-based AmB intravenous medications available on the market are authorised for treating and preventing pulmonary fungal infections. Nevertheless, these formulations are not intended for pulmonary administration, necessitating patient hospitalisation to properly reconstitute the medications by trained personnel and administer them via a suitable nebuliser apparatus [1]. Lung delivery results in higher local lung concentrations, decreasing the overall dose needed and limiting the exposure of the drugs in other areas, such as the kidney, to limit their nephrotoxicity.

Previously, a dry powder inhaler (DPI) formulation for lung delivery of AmB was successfully developed [1]. However, there is an increasing trend in AmB-resistant isolates of *A. fumigatus* [11,12]; hence, the use of fixed-dose combination products can be a useful approach to overcome this challenge [13,14,15]. Based on this premise, the pulmonary co-delivery of AmB with itraconazole (ITR) can be an interesting strategy when considering their complementary mechanisms of action [16]. ITR is an antimycotic drug that belongs to the azoles group. This molecule blocks the ergosterol synthesis through a different mechanism of action, in which the inhibition of the lanosterol 14α-demethylase takes place [16,17]. Currently, ITR formulations are only available on the market for intravenous and oral administration, although a few attempts to achieve a suitable pulmonary delivery of ITR have also been investigated [18].

Pulmonary administration can be carried out using different types of inhalers, including pressurised metered-dose inhalers (pMDIs) or DPIs [1,19]. DPIs have two major advantages over pMDIs. In relation to DPIs, their use of a propellant and the coordination between the device actuation and the patient inhalation are not required, providing a more efficient deposition of the powder in the lungs and increasing the effectiveness of the formulation with a single dose [1,19]. However, the efficiency of inhalation is dependent on the inspiratory flow rate of the patient, which might hinder drug delivery in patients with a decreased pulmonary capacity [19]. Nebulisers can also be utilised to administer medicines through the pulmonary pathway; however, in this case, the medicine needs to be in the form of a liquid solution that will turn into a mist for inhalation. Nonetheless, this can result in drug degradation, considering the lack of AmB stability in aqueous media [7]. Also, nebulisation requires a more lengthy administration process and the utilisation of a mouthpiece that is tightly sealed with the lips of the patient, and is associated with overall poorer patient compliance [20].

The hypothesis underpinning this work is that creating a formulation for pulmonary administration to simultaneously co-deliver AmB and ITR could effectively target the site of action for treating and preventing severe fungal lung infections, such as aspergillosis. This dual formulation, containing both AmB and ITR, has the potential to significantly improve upon the current off-label options for pulmonary administration. Consequently, the primary goal of this research was to develop a DPI consisting of microparticles (MPs) that can efficiently deliver AmB and ITR throughout all regions of the respiratory tract, which is crucial for fully eradicating fungal pulmonary infections. To achieve this, a Design of Experiment (DoE) approach was conducted, aiming to optimise the particle size and aerosolization characteristics of the formulation, thereby ensuring optimal lung deposition. A comprehensive physicochemical characterisation was carried out, including scanning electron microscopy (SEM) to examine morphology, differential scanning calorimetry (DSC) to understand crystallinity, and mass median aerodynamic diameter (MMAD) and fine particle fraction (FPF) to assess lung deposition.

## 2. Materials and Methods

### 2.1. Materials

AmB was purchased from North China Pharmaceutical Huasheng Co. (Hebei, China). ITR was purchased from Kemprotec Ltd. (Camforth, UK). γ-CD was a sample gift kindly provided by Ashland (Madrid, Spain). Mannitol was supplied from Roquette (Lestren, France), while leucine was purchased from Sigma-Aldrich (Dublin, Ireland). Phosphoric acid (85%) (H_3_PO_4_) and sodium hydroxide (NaOH) were purchased from Panreac S.A. (Barcelona, Spain). All solvents utilised were HPLC grade unless otherwise stated.

### 2.2. Methods

#### 2.2.1. Manufacturing of MPs Containing AmB and ITR (AmB-ITR MPs)

The manufacturing process for the formulations consisted of first encapsulating AmB within γ-CD and, subsequently, adding the remaining excipients and ITR before undergoing spray drying [21]. For this purpose, γ-CD was dissolved in 50 mL of deionised water and the pH was adjusted to pH 12 using NaOH 1 N, followed by the addition of AmB. The solution was then neutralised to pH 7.4 using H_3_PO_4_ (10% *v*/*v*) and, after that, mannitol, leucine, and ITR were added. The amount of each component in the formulation was determined with a DoE approach (see Section 2.2.2), except for mannitol. The combined mass of the mixture was 500 mg, with the quantity of mannitol in each formulation contingent upon the other components. The mixture was then vortexed and placed in an ultrasonic bath for 10 min at room temperature. Then, the formulation was probe sonicated at 10 W for 10 s using a Branson Sonifier 250 (Emerson Industrial, Brookfield, CT, USA) to obtain a homogenous suspension. A Büchi Mini Spray-Dryer in the open mode was used to spray dry the resultant solution (Büchi Labortechnik, Flawil, Switzerland). It utilised air as the atomizing gas using a two-fluid nozzle with a 1.5 mm cap and a 0.7 mm nozzle tip. Process parameters were adjusted according to DoE specifications. AmB-ITR MPs were collected from the collector compartment vessel of the spray dryer, and samples were stored at desiccated conditions at 5 ± 3 °C until further analysis was performed.

#### 2.2.2. Optimisation of AmB-ITR MPs

##### Taguchi Design

The optimisation of the composition of AmB-ITR MPs was carried out using DoE in two iterations with Design-Expert^®^ version 10 software (Stat-Ease^®^, Minneapolis, MN, USA). First, a seven-factor eight-run Taguchi design with two levels (2^7^) allowed a pre-screening identification of the formulation and process parameters with a high influence on the quality of the final product [22,23]. All independent variables (factors) in the Taguchi design were numerical, including the amount of AmB (5 or 10%), the amount of ITR (0 or 10%), the amount of γ-CD (5 or 15%), the amount of leucine (5 or 15%), the inlet temperature in the spray dryer (120 or 150 °C), the solution feed rate (5 or 10 mL/min), and the gas flow rate (600 or 800 NL/h). Meanwhile, five dependent variables (responses) were investigated, including the yield of the process, particle size distribution, in vitro efficacy against *C. albicans*, AmB encapsulation efficiency, and AmB aggregation state, as described below.

Determination of the yield

To calculate the yield of the process, the product was weighed after collection and compared to the expected amount of product (Equation (1)):(1)$$Yield(\%)=\frac{A}{B}\times 100$$
where *A* is the weight of the collected product and *B* is the total weight of the spray-dried solute/solid material.

Determination of particle size

Laser diffraction was used to measured particle size in triplicate using a Mastersizer 3000 (Malvern Panalytical Ltd., Malvern, UK). A Malvern Aero S. dry powder feeder was used at a pressure of 1 bar and a vibration feed rate of 50%. Results were expressed as median particle size (D_50_) in µm.

Determination of the antifungal in vitro activity

In vitro antifungal activity was assessed with agar diffusion assay, according to the European Pharmacopoeia standards as previously described [23,24,25]. For this purpose, *C. albicans* CECT 1394 (Universidad de Alcalá, Madrid, Spain) was used. After incubation at 30 °C for 72 h, yeasts were initially isolated in a Petri dish using Sabouraud dextrose agar (Becton Dickinson and Co., Franklin Lakes, NJ, USA). Then, using the McFarland factor, a yeast suspension was created in sterile NaCl 0.9%. Two hundred millilitres of Müller–Hinton agar (Laboratorios Conda S.A., Madrid, Spain), supplemented with glucose (Panreac Química S.A.U., Barcelona, Spain) (2% *w*/*v*), were autoclaved (121 °C, 20 min) to ensure the assay took place in aseptic conditions. Sterile agar was then supplemented with 30 µL of an aqueous solution of methylene blue (0.5 mg/mL) to enhance the contrast when reading the inhibition halos. Once the Müller–Hinton agar cooled down (~40 °C), *C. albicans* was inoculated. Inoculated agar was then placed onto Petri dishes and, when it solidified, paper disks (6 mm) were embedded with 20 µL of AmB-ITR formulation (0.5 mg/mL) and were placed on top. Four standards were prepared for comparison purposes at the same concentration, including AmB in water, AmB in dimethyl sulfoxide (DMSO) (Scharlab S.L., Barcelona, Spain), ITR in water, and ITR in DMSO. Plates were then incubated at 30 °C for 48 h and the diameter of the inhibition halo was measured. Results were expressed as inhibition halo in mm.

Determination of the encapsulation efficiency

Encapsulation efficiency was measured by adding 2 mL of DMSO to 5 mg of spray-dried powder. A 0.45 µm PTFE filter was used for filtering the samples (Thermo Fisher Scientific Inc., Waltham, MA, USA), which were diluted for high-performance liquid chromatography analysis (HPLC), using a previously validated method for AmB and ITR [1]. Analysis was performed in a Jasco HPLC system (Jasco Co., Tokyo, Japan) with an AS-2050 Plus intelligent autosampler, an LG-2080-02 ternary gradient unit, a DG-2080-53 3-line degasser, a PU-1580 intelligent HPLC pump and a UV-1575 intelligent UV-vis detector. The mobile phase for both AmB and ITR consisted of a mixture of acetonitrile/acetic acid/water (52:4.3:3.7 V:V:V) and was filtered through a 0.45 µm Supor^®^-450 membrane (Pall Co., Bloomfield Township, MI, USA). A BDS Hypersil C18, 5 µm (200 × 4.6 mm) column (Thermo Fisher Scientific Inc., Waltham, MA, USA) was utilised for separation, and the UV detection wavelength was set at 406 and 260 nm for AmB and ITR, respectively, while the injection volume was 100 µL. Each sample was quantified in two different runs, setting the corresponding wavelength for each drug. The retention time for AmB was 8 min, while it was 20 min for the ITR. Elution took place in an isocratic method at room temperature with a flow rate of 1 mL/min. Encapsulation efficiency was then calculated and expressed as a percentage (Equation (2)):(2)$$Encapsulation\;efficiency\;(\%)=\frac{A}{B}\times 100$$
where *A* is the amount of drug recovered and *B* is the total amount of drug initially included in the formulation, which was considered as 100%. The encapsulation efficiency was calculated for AmB and ITR independently.

Determination of the AmB aggregation state

Finally, a relationship between both aggregation states of AmB was established (Equation (3)). AmB is a molecule with a high tendency for aggregation in aqueous media and can coexist in three different aggregation states (monomeric, dimeric (oligomeric), and polyaggregated states) with different efficacy and toxicity profiles [26]. For this purpose, samples were diluted in deionised water up to 10 µg/mL; then, absorbance at a wavelength of 331 and 408 nm was measured using a Jasco V-730 spectrophotometer (Jasco Co., Tokyo, Japan) to identify the dimeric and monomeric aggregation states of AmB, respectively [1].
(3)$$\frac{dimer}{monomer}ratio=\frac{{Abs}_{331nm}}{{Abs}_{408nm}}$$

##### Box–Behnken DoE

A second DoE iteration was carried out once the critical quality attributes (CQAs) were identified. In this case, a three-factor three-level Box–Behnken design was performed to optimise the final formulations [27]. The three selected factors were the total amount of both drugs (AmB and ITR) (20, 30, and 40%), the amount of leucine (10, 15, and 20%), and the gas flow rate (600, 700, and 800 NL/h). The same dependent variables were investigated as described above, except for in vitro antifungal activity (i.e., antifungal efficacy), for which the result was expressed as a percentage of that obtained with an AmB standard, rather than the size of the inhibition halo.

In order to identify a mathematical connection between the responses, DesignExpert^®^ enabled the computation of the Bonferroni and t-value limits in the Taguchi design, as well as the creation of multiple linear regression (MRL models). In order to understand the relevance of the various components in the spray-drying responses, analysis of variance (ANOVA) was conducted. All dependent variable values were predicted for all runs by inserting the values of the numerical factors. The best formulation from the Box–Behnken design was chosen using the numerical optimisation method.

#### 2.2.3. Physicochemical Characterisation of Optimised AmB-ITR MPs

##### Scanning Electron Microscopy (SEM)

The morphology of AmB-ITR MPs was examined using SEM. For this purpose, samples were first placed onto carbon tabs, mounted on aluminium pins, and sputtered with gold. Gold-coated samples were then observed using a JEOL JSM-6335F microscope (JEOL Ltd., Tokyo, Japan) at an accelerated voltage of 15 kV. Elemental analysis of the MPs was also carried out by energy-dispersive X-ray spectroscopy (EDX) using an Oxford Instruments X-Max (Oxford Instruments, Abingdon, UK) to understand the localisation of ITR within the MPs. For this purpose, an elemental analysis of chlorine was performed.

##### Modulated Temperature Differential Scanning Calorimetry (MT-DSC)

A DSC Q200 (TA Instruments^®^, New Castle, DE, USA) was utilised with nitrogen as the purge gas. A temperature range of 0–200 °C was used to scan the formulation (2–3 mg) previously placed in aluminium pans with a modulation rate of 0.8 °C every 60 s at 5 °C/min [16]. Thermograms of AmB and ITR were also acquired for comparison purposes. The glass transition of ITR was investigated by melt quenching the sample. In the latter, ITR was heated at a rate of 10 °C/min until 200 °C, then cooled down to 25 °C, and followed the same modulation temperature protocol as above described. TA Universal Analysis^®^ software version 4.5A (TA Instruments^®^, New Castle, DE, USA) was used to process the data. Indium was used as the standard during the calibration of the instrument. The percentage of ITR crystalline was calculated with consideration of the enthalpy of fusion of unprocessed ITR and the amount of ITR in the sample.

#### 2.2.4. In vitro Lung Deposition

To assess the in vitro deposition of AmB-ITR MPs, mass median aerodynamic diameter (MMAD) and fine particle fraction (FPF) (<3 and 5 µm) were calculated, as previously described [1,28]. A Next Generation Impactor™ (NGI™; MSP Corporation, Shoreview, MN, USA) was used, which was equipped with seven stainless compartments (stages), a stainless steel induction port and one micro-orifice collector (MOC). To prevent particle bouncing and facilitate sample collection, filter paper embedded with 1 mL of a mixture of acetonitrile and water (1:1, V:V) was placed in each stage. An air flow of 60 L/min was adjusted using a Copley TPK 2000 critical flow controller, which was connected to the NGI™ apparatus, and a Copley HCP5 vacuum pump (Copley Scientific, Colwick, UK). Inhalation time was set at 4 s, with a total of 4 litres of inhaled air. A hydroxypropyl methylcellulose capsule (No. 3) filled with 25 ± 2 mg of formulation (*n* = 3) was placed in a HandiHaler^®^ inhaler (Boehringer Ingelheim, Ingelheim, Germany) for inhalation, facilitating the deposition of MPs at different stages of the NGI™ apparatus. The content of AmB and ITR was determined using the previously validated HPLC method [1]. Aerodynamic cut-off diameters (ACD) at the flow rate used were as follows: 8.06 µm, 4.46 µm, 2.82 µm, 1.66 µm, 0.94 µm, 0.55 µm, 0.34 µm, and <0.34 µm from stage 1 to stage 8, respectively [29]. Data analysis software from Copley Scientific (Nottingham, UK) was used for the MMAD analysis.

#### 2.2.5. Statistical Analysis

Minitab^®^ 21 (Minitab Inc., Coventry, UK) was employed to conduct one-way ANOVA. Tukey’s test was applied to compare the formulations and determine any statistically significant differences between the groups.

## 3. Results

### 3.1. Optimisation of AmB-ITR-MPs

#### 3.1.1. Taguchi Design

The factors and responses from the Taguchi design are shown in Table 1. Pareto charts were constructed to understand the influence of each factor on each dependent variable (Figure 1). At the tested conditions, the amount of leucine and inlet temperature had a significant positive effect on the yield, while the solution feed rate showed a negative effect. However, although the gas flow rate and the amount of ITR showed a positive effect on the yield, while the amounts of γ-CD and AmB exhibited a negative effect, these factors were not significant (Figure 1a). Both the solution feed rate and the amount of ITR promoted a significant increase in particle size, while the inlet temperature, the amount of leucine, and the gas flow rate had a significant impact on particle size, promoting a smaller particle size. The amount of AmB promoted an increase of particle size and the amount of γ-CD enhanced an opposite effect; nevertheless, the impact was not significant (Figure 1b). Regarding in vitro antifungal efficacy, AmB-ITR MPs were more effective at a high gas flow rate and when a high amount of leucine and a high solution feed rate were used, while the factors with a negative impact on antifungal efficacy were the amount of γ-CD and the gas flow rate (Figure 1c). Nevertheless, all formulations showed effectiveness against *C. albicans* (inhibition halo > 15 mm) (Table 1) [30]. The amount of AmB did not exhibit an increase in effectiveness, while the amount of ITR did, although these effects were not significant. The encapsulation efficiency was positively influenced by the amount of γ-CD and inlet temperature. The positive effect of γ-CD might be explained by the high entrapment capacity of the AmB within γ-CD. However, the amount of ITR and AmB had a negative impact on the encapsulation efficiency, as higher amounts of the drug might be competing for entrapment within the γ-CD. The amount of leucine and the gas flow rate also exhibited a negative effect in encapsulation efficiency, although the effect was not significant in these cases (Figure 1d). Finally, the amount of AmB and leucine triggered the dimeric aggregation state of AmB with a significant effect, while the dimerization triggered by the amount of ITR and the solution feed rate was not significant. On the other hand, the inlet temperature, the amount of γ-CD, and the gas flow rate promoted a monomeric AmB, but without a significant effect (Figure 1e).

#### 3.1.2. Box–Behnken Design

Based on the results from the Taguchi design, a Box–Behnken design with 17 runs was constructed. Only four variables were further investigated, including the amount of both drugs (AmB and ITR), the amount of leucine, and the gas flow rate. These four factors were selected with particle size taking priority over other variables, although other factors might have shown a significant impact (e.g., inlet temperature showed a significant effect on the yield), as particle size is critical for lung deposition. A smaller particle size is obtained at a higher inlet temperature, as the powder is dried more intensely. For this reason, we decided to keep the highest testing level at a constant of 150 °C. The amount of γ-CD was adjusted based on the total amount of the drug added to the formulation in a 1:1 ratio (w:w). The inlet temperature was kept at 150 °C, and the solution feed rate was kept at 5 mL/min. Independent and dependent variables for the 17 runs of the Box–Behnken design are displayed in Table 2.

The gas flow rate had a positive impact on the yield, i.e., the higher the gas flow rate, the higher the yield (Figure 2a–c). Additionally, the yield was maximised when a higher amount of drug (both AmB and ITR) and a low amount of leucine were combined. Particle size was minimised when a higher gas flow rate was used in combination with a higher amount of AmB and ITR (Figure 2d). Finally, the aggregation state was primarily dependent on the amount of drug. The higher the amount of spray-dried AmB and ITR, the higher the aggregation state of AmB toward a dimeric state (Figure 2e). The other two variables, AmB encapsulation efficiency and in vitro efficacy, did not show significant differences (*p* > 0.05) for the three factors evaluated (*p* > 0.05). Statistical analyses and mathematical equations are included in the Appendix A section.

In total, three optimised formulations were engineered based on three different amounts of total antimicrobial drugs (20, 30, and 40%), aiming to minimise particle size and maximise the yield. The results of each dependent variable, as well as the final composition, are shown in Table 3. Seventy percent of the responses were within the confidence interval predicted with the multilinear regression models. F1 showed the best results in terms of yield, particle size, and efficacy. The three formulations exhibited an aggregation ratio (dimer/monomer) below 1, suggesting that the AmB monomeric state was more prevalent; however, it is expected that an equilibrium with the dimeric state occurs in physiological conditions [31]

### 3.2. Physicochemical Characterisation of AmB-ITR MPs

#### 3.2.1. Scanning Electron Microscopy (SEM)

SEM micrographs revealed that the MPs possessed an irregular non-smooth collapsed surface, as depicted in Figure 3. Unencapsulated crystals were also observed in the micrographs. Evidence of chlorine was found on the crystals, indicating that ITR might be partially encapsulated. The number of crystals appeared to be increased at higher loadings of antimicrobials (F3). EDX spectra exhibiting the presence of chlorine can be found in the Appendix A.

#### 3.2.2. Modulated Temperature Differential Scanning Calorimetry (MT-DSC)

DSC thermograms exhibited a sharp endothermic peak corresponding to the ITR melting point at 163.24 °C. A depression in the melting point was observed for F1, F2, and F3 to 152.29 °C, 159.97 °C, and 160.19 °C, respectively. The percentage of crystalline ITR was 5.79, 61, and 80% for F1, F2 and F3, respectively, which aligns with the results of the SEM micrographs. A higher density of crystals is observed when a higher percentage of both antifungal drugs was used. After melt quenching, the ITR showed a glass transition temperature (Tg) at 56.34 °C. However, this Tg was not observed in any of the formulations (Figure 4). A broad endothermic event was observed for the unprocessed AmB at 105.87 °C, which shifted to lower temperatures (83.51 °C, 82.11 °C and 95.40 °C) for F1, F2, and F3, respectively. Later, an endothermic event for AmB was not observed as it degraded before melting (Figure 4).

### 3.3. Mass Median Aerodynamic Diameter (MMAD) and Fine Particle Fraction (FPF)

Results of the in vitro deposition characteristics of AmB-ITR MPs are shown in Table 4. In all cases, AmB showed higher FPF values, both below 5 and 3 µm, compared to ITR. The best in vitro deposition was observed for F1 and F2. F3 showed poorer deposition characteristics, which can be related to the larger unencapsulated ITR fraction compared to F1 and F2.

MMAD for AmB was found to be in the optimal range for pulmonary administration in all cases (MMAD ~ 3 µm) (Figure 5). Deposition in the first stage of the NGI™ apparatus was minimal (<5%), indicating a deeper lung deposition pattern for AmB compared to ITR. The percentage of particles deposited in the region between 4.46 and 1.66 µm was the highest for AmB, although a small fraction (5–10%) reached the later stages, which is indicative of the deepest region of the respiratory tract (Figure 5a,b). On the other hand, ITR showed significantly higher values of MMAD, with lower values of FPF below 5 and 3 µm compared to AmB, indicating a pulmonary delivery which is limited to the upper respiratory tract. ITR was primarily found in the first two stages of the NGI™ apparatus, while less than 20% of the total dose accumulated in the stages was representative of the deeper regions of the respiratory system (Figure 5c,d). This was most noticeable for the F3 formulation, which can be related to the greater amount of unencapsulated ITR crystals. In all the cases, the FPF is calculated for each drug independently. Based on the results obtained, AmB and ITR-loaded microparticles have a suitable lung deposition pattern in the deeper regions of the respiratory tract. Even though there is a significant unencapsulated ITR fraction corresponding to the crystals observed in SEM, it still possesses a particle size small enough to reach the lung parenchyma within the upper regions.

## 4. Discussion

To the best of our knowledge, this is the first report of a formulation designed for pulmonary administration that combines AmB with another antifungal drug to enhance the antimicrobial effect. Three formulations containing a high percentage of AmB and ITR (20–30–40%) and combined in equal amounts were optimised using DoE tools and produced using spray drying, which is an easily scalable industrial manufacturing method [32,33,34]. All the excipients used were carefully selected to ensure that they were generally regarded as safe (GRAS) [35]. γ-CD stands out with the largest cavity size among natural cyclodextrins, making it highly suitable for accommodating large molecules, such as AmB [31]. Mannitol and leucine were chosen as the spray-drying carrier constituents due to their unique characteristics of enabling the preparation of MPs with varying surface roughness and improving lung deposition [36,37,38]. Leucine was also incorporated to protect the formulations against moisture-induced changes [39]. Since none of the investigated variables demonstrated a significant effect on antifungal efficacy, three formulations were developed using the optimised excipient composition. The modification involved increasing drug amounts from 20% and 30% up to 40% of antifungals in a 1:1 AmB:ITR weight ratio.

SEM images verified that the formulations possessed a rough surface, exhibiting an irregular and non-spherical shape, likely attributable to the incorporation of mannitol and leucine during the spray-drying process. EDX evidenced that the free crystals corresponded to unencapsulated ITR, due to the presence of chlorine (Appendix A). ITR is the only molecule that has Cl- groups in its structure. If ITR would be fully encapsulated within the microparticles, Cl- would not have been detected with EDX. No chlorine was detected on the surface of the microparticles, while it was found on the elongated crystals visualized with SEM. The marketed oral solution of itraconazole (Sporanox^®^) contains 40% of 2-hydroxypropyl-β-cyclodextrin [40]. Β-CD has a smaller cavity than γ-CD, which indicates that the latter is large enough to accommodate the ITR molecules; however, the amount of CD required to solubilise ITR is high. During the formulation process, AmB is first incorporated in the aqueous medium, which can trigger the initial complexation between AmB and CD. ITR is added in a later step so only the free uncomplexed γ-CD is available for interaction. This results in a poorer solubilisation capacity and the failure to encapsulate all the ITR content, which is more evident when larger amounts of ITR are used as AmB displaces ITR from its interaction with the CD, resulting in a significantly higher percentage of crystalline ITR at a higher total content of antimicrobials. SEM micrographs showed a higher density of free ITR crystals in F3, which was the formulation with the highest antimicrobial content. This aligns with the higher crystalline ITR content observed in the DSC thermograms and the absence of a Tg, especially in the F3 formulation [41].

The in vitro deposition profile differed significantly between both antimicrobials. The smaller MMAD for AmB correlates with an expected deeper lung deposition, while ITR showed a significantly larger MMAD and a deposition corresponding primarily to the upper parts of the respiratory tract. The deposition pattern for AmB is close to the one previously reported in [1], although the value for FPF below 3 µm was lower, which can be related to the greater percentage of AmB in the formulation, from 5 up to 10–15–20% content (2–4 fold greater). This is a potential pharmacological advantage, as the formulation could elicit a greater pharmacological effect, considering the limited dose that can be delivered with each administration of a DPI. The complementary distribution in lung deposition between AmB and ITR can be useful, bearing in mind that *Aspergillus* spp. can cause upper and lower airway infections [42].

Due to the poor aqueous solubility of ITR, previous pulmonary delivery studies have focused on its encapsulation within a nanotransferosomal formulation to enhance drug solubilisation [43]. In this previous study, the nanotransfersome particle size was about 500 nm and, after co-spray drying with mannitol, the resulting microparticles ranged between 1 and 5 µm. Nevertheless, the in vitro deposition pattern was similar to the results obtained in this work, with an MMAD above 5 µm and a deposition primarily in the upper region of the respiratory tract. Also, the encapsulation in transferosomes required the use of surfactants, such as Span^®^ or ursodiol, which can cause adverse pulmonary effects, and the drug loading was below 5%, which would necessitate more administration to achieve the same delivered dose of ITR.

Overall, the development of a DPI formulation for the co-delivery of AmB and ITR to the lungs was successful, considering that all the upper and lower pulmonary regions were exposed to high concentrations of antifungals. This is a key point, considering that *Aspergillus* spp. necessitates higher doses of antifungals to be eradicated compared to *Candida* spp. [44,45]. Also, the thermodynamical properties of the particles should be investigated in more detail to understand the interaction between the pulmonary surfactant model monolayer and the different doses which could be utilized [46]. To reduce the unencapsulated ITR fraction, larger amounts of CD could be used. However, this excipient has still not been approved for pulmonary delivery and it is currently under clinical trials (www.clinicaltrials.gov; accessed on 9 September 2023) [47]. For this reason, the amount of CD used was reduced to the minimum. Further pharmacokinetic studies should be performed to evaluate the residence time of the drugs in the lung.

## 5. Conclusions

Co-delivering a high dose of AmB and ITR via a DPI for pulmonary administration presents a viable alternative for the treatment of severe lung infections, such as aspergillosis. This approach enables the direct targeting of antifungals to the site of action, thereby avoiding undesired systemic effects, such as nephrotoxicity in the case of AmB. The engineered formulations exhibited a complementary in vitro deposition pattern, corresponding to an upper airway ITR distribution along with a lower airway AmB deposition. This offers significant advantages for treating fungal infections not only in the lung parenchyma but also in the upper respiratory tract. The deposition profile of ITR, which indicates that it will be limited to the upper regions of the respiratory tract, is related to the unencapsulated crystalline fraction which has poorer aerosolization characteristics, which are presumably related to a poorer flow performance. The use of a larger quantity of CD can resolve this pharmacotechnical issue.

## Figures and Tables

**Figure 1 pharmaceutics-15-02601-f001:**
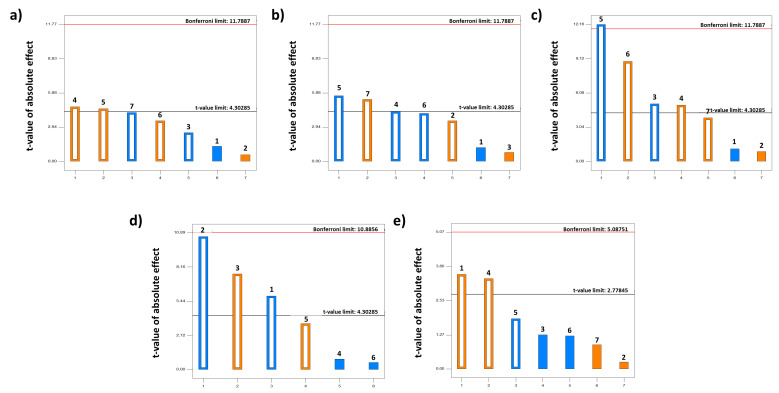
Pareto charts describing the influence of the seven factors in the Taguchi design: (**a**) yield; (**b**) particle size; (**c**) inhibition halo; (**d**) encapsulation efficiency; (**e**) aggregation rate. The orange bar represents a positive effect, while the blue bar represents a negative effect. Key: (1) amount of AmB, (2) amount of ITR, (3) amount of γ-CD, (4) amount of leucine, (5) inlet temperature, (6) gas flow rate, (7) solution feed rate. Bonferroni limit: effects that are above the Bonferroni limit are almost certainly important; t-value limit: effects that are above the t-value limit are possibly important.

**Figure 2 pharmaceutics-15-02601-f002:**
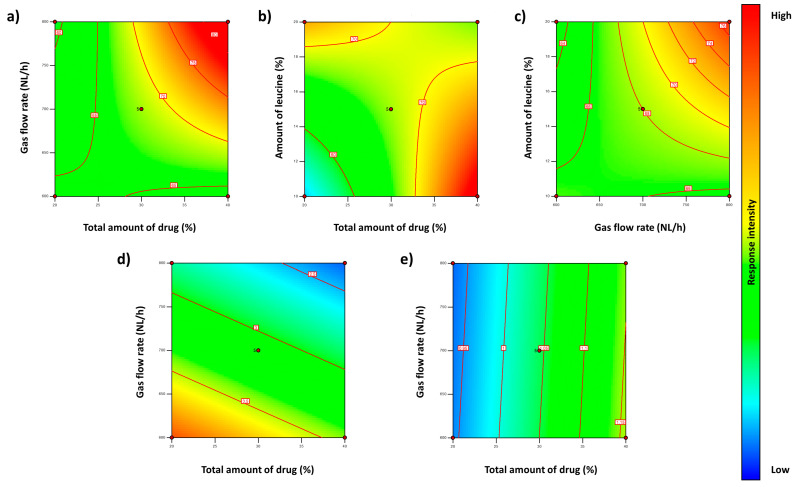
Two-dimensional contour plots showing the influence of the total amount of drug, the gas flow rate, and the amount of leucine on yield (**a**–**c**), particle size (**d**), and aggregation ratio (**e**).

**Figure 3 pharmaceutics-15-02601-f003:**
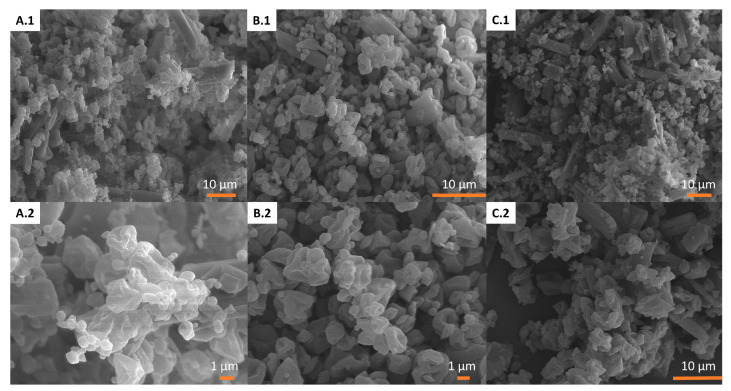
SEM micrographs of F1 (**A.1**,**A.2**), F2 (**B.1**,**B.2**), and F3 (**C.1**,**C.2**) at different magnifications.

**Figure 4 pharmaceutics-15-02601-f004:**
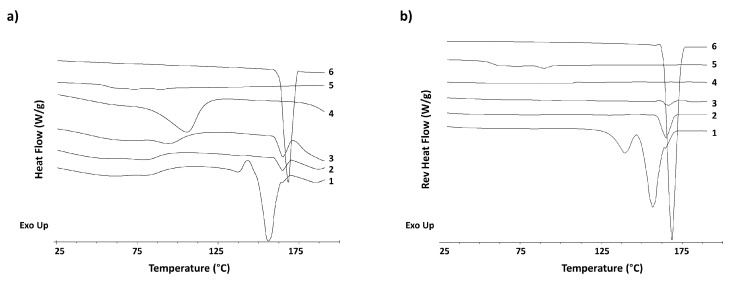
Physicochemical characterisation of AmB-ITR-MPs: (**a**) DSC thermograms showing heat flow; (**b**) DSC thermograms showing reverse heat flow. Key: F1 (1), F2 (2), F3 (3), AmB raw material (4), ITR melt quench (5), ITR raw material (6).

**Figure 5 pharmaceutics-15-02601-f005:**
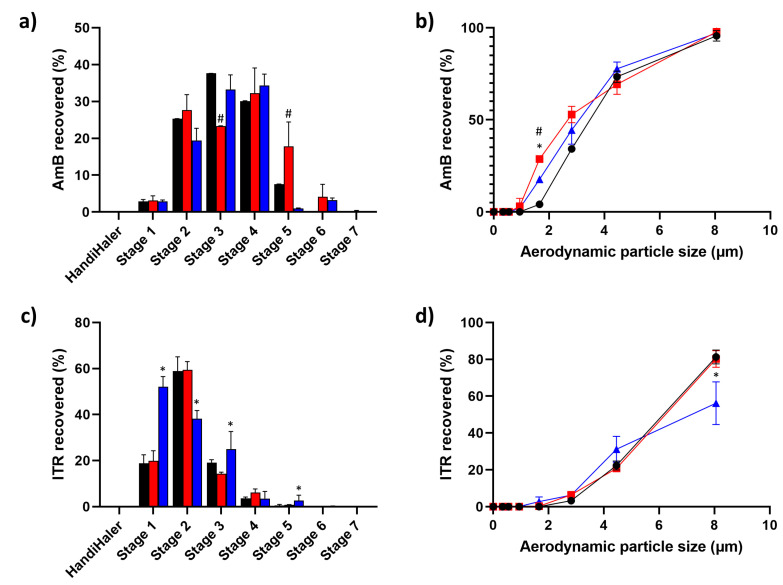
In vitro deposition of F1, F2, and F3 tested in NGI™ apparatus: (**a**) percentage of AmB recovered at different stages; (**b**) cumulative AmB deposition; (**c**) percentage of ITR recovered at different stages; (**d**) cumulative ITR deposition. Key: F1 (black), F2 (red), F3 (blue). Statistically significant differences are represented by # (F2 vs. F1 and F3) and * (F3 vs. F1 and F2) (*p* < 0.05, one-way ANOVA post hoc test).

**Table 1 pharmaceutics-15-02601-t001:** Pre-screening Taguchi design matrix for AmB-ITR MPs.

Run	Factor 1	Factor 2	Factor 3	Factor 4	Factor 5	Factor 6	Factor 7	Response 1	Response 2	Response 3	Response 4	Response 5
	AmB (%)	ITR (%)	γ-CD (%)	Leucine (%)	Inlet Temperature (°C)	Solution Feed Rate (%)	Gas Flow Rate (NL/h)	Yield (%)	Particle Size, D_50_ (µm)	Inhibition Halo (mm)	AmB Encapsulation Efficiency (%)	Aggregation Ratio (Dimer/Monomer)
1	5	0	5	5	120	5	400	26.56	4.87	25.12	76.97	0.72
2	10	10	5	15	120	10	400	25.46	5.76	26.62	187.28	1.33
3	10	10	5	5	150	5	800	64.92	2.86	24.51	53.77	0.77
4	10	0	10	15	120	5	800	49.80	2.49	26.45	77.01	1.05
5	5	10	10	5	120	10	800	12.96	6.33	26.60	70.73	0.62
6	5	0	5	15	150	10	800	71.32	2.52	26.09	80.48	0.81
7	10	0	10	5	150	10	400	8.58	4.67	22.60	86.40	0.82
8	5	10	10	15	150	5	400	65.00	3.34	23.14	76.61	0.76

**Table 2 pharmaceutics-15-02601-t002:** Independent and dependent variables according to the Box–Behnken design. Factor 1 includes AmB and ITR in a 50:50 (w:w) ratio.

Run	Factor 1	Factor 2	Factor 3	Response 1	Response 2	Response 3	Response 4	Response 5	Response 6
	Total Amount of Drug (%)	Gas Flow Rate (NL/h)	Amount of Leucine (%)	Yield (%)	Particle Size, D_50_ (µm)	Efficacy (%)	AmB Encapsulation Efficiency (%)	ITR Encapsulation Efficiency (%)	Aggregation Ratio (Dimer/Monomer)
**1**	20	700	10	44.56	4.08	94.19	65.18	86.39	0.892
**2**	30	700	15	74.72	3.29	91.81	86.97	95.94	0.939
**3**	40	700	10	74.48	3.16	93.75	55.95	96.31	1.161
**4**	30	800	20	72.68	2.21	92.38	88.62	87.01	1.120
**5**	40	700	20	71.22	2.44	91.61	100.27	92.01	1.183
**6**	20	600	15	62.22	3.63	98.75	91.86	92.01	1.131
**7**	30	700	15	68.14	2.40	96.89	87.95	95.15	1.088
**8**	30	600	20	57.70	4.01	91.09	79.01	72.52	0.994
**9**	20	700	20	75.82	2.99	93.06	80.65	65.12	0.907
**10**	40	800	15	77.32	2.56	93.30	79.63	75.63	1.120
**11**	40	600	15	58.84	3.60	98.46	92.06	99.68	1.270
**12**	30	800	10	71.34	2.15	91.13	84.16	100.63	1.136
**13**	30	700	15	78.46	3.35	86.09	89.88	100	0.977
**14**	30	600	10	71.22	3.15	88.67	93.33	83.18	0.972
**15**	30	700	15	71.28	3.24	95.76	94.20	100	1.011
**16**	30	700	15	71.96	3.82	88.14	94.53	93.37	0.910
**17**	20	800	15	54.22	3.02	90.56	88.35	71.82	0.945

**Table 3 pharmaceutics-15-02601-t003:** Composition and dependent variables of three optimised formulations of AmB-ITR MPs. Key: confidence intervals are shown in brackets. The total amount of drug involves AmB and ITR in a 50:50 (w:w) ratio.

Formulation	Total Amount of Drug (%)	Gas Flow Rate (NL/h)	Amount of Leucine (%)	Amount of γ-CD (%)	Amount of Mannitol (%)	Yield (%)	Particle Size, D_50_ (µm)	Efficacy (%)	AmB Encapsulation Efficiency (%)	ITR Encapsulation Efficiency (%)	Aggregation Ratio (Dimer/Monomer)
**F1**	20	800	20	20	40	78.54 (57.26–88.34)	3.05 (1.98–3.05)	100 (90.89–94.47)	82.66 (78.25–96.30)	82.81 (85.49–107.12)	0.924 (0.87–1.05)
**F2**	30	768	20	30	20	69.56 (65.24–82.84)	3.19 (2.08–2.91)	94.89 (90.89–94.47)	88.14 (81.08–91.63)	93.06 (90.16–101.38)	0.909 (0.990–1.120)
**F3**	40	667	10	40	10	69.20 (66.02–88.62)	3.45 (2.89–3.87)	90.00 (90.89–94.47)	87.82 (77.45–106.80)	83.62 (90.73–103.30)	0.911 (1.03–1.19)

**Table 4 pharmaceutics-15-02601-t004:** FPF below 5 and 3 µm for F1, F2, and F3 at 60 L/min for 4 s.

Formulation	Drug	FPF < 5 µm (%)	FPF < 3 µm (%)	MMAD (µm)
F1	AmB	81.60 ± 0.05	43.07 ± 0.04	3.39 ± 0.01
	ITR	42.24 ± 11.33	12.91 ± 3.01	5.80 ± 0.07
F2	AmB	91.17 ± 1.14	43.48 ± 4.29	3.03 ± 0.07
	ITR	25.78 ± 0.64	8.43 ± 0.01	5.77 ± 0.31
F3	AmB	67.59 ± 8.12	35.71 ± 8.09	3.04 ± 0.35
	ITR	38.63 ± 9.53	15.15 ± 4.99	6.79 ± 0.27

## Data Availability

Data unavailable due to privacy restrictions.

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
