# Peer review of "Co-Delivery of a High Dose of Amphotericin B and Itraconazole by Means of a Dry Powder Inhaler Formulation for the Treatment of Severe Fungal Pulmonary Infections"

_pharmaceutics, 2023, doi:10.3390/pharmaceutics15112601_

Round 1
Reviewer 1 Report
Comments and Suggestions for Authors
In this study, the authors tailored microparticles containing a high dose high dose of amphotericin B and itraconazole for inhalation in an attempt to manage severe fungal pulmonary infections. The idea of the work is interesting. However, the manuscript could not be accepted in the present form for the following points
1) An HPLC chromatogram for simultaneous determination of AmB and ITR should be provided.
2) The description of results in Section 3.1. need to be revised regarding the impact (positive/negative effects) of different formulation variables on microparticle characteristics since Pareto charts did not support their explanation.
3) On what basis the authors selected the amount of both drugs (AmB and ITR), the amount of leucine, and the gas flow rate. As formulation variables for further Box Behnken design given that Factor such as inlet temperature which had certain impact on yield was not included.
4) In Box Behnen design, statistical analysis data (ANOVA) along with quadratic equations must be provided to better explain the impact of different formulation variables on product responses
5) In section 3.2.1., the authors claimed that “Evidence of chlorine was found on the crystals indicating that ITR might be partially encapsulated”. How the authors confirm this claim.
6) Is irregular non-smooth surface microparticles are suitable for inhalation and lung deposition.
7) In section 3.2.2., what the authors means by “The percentage of crystalline ITR was 5.79, 61, and 80% for F1, F2 and F3, respectively, which align with the SEM micrograph results.”
- How the authors calculated the percentage of crystalline ITR in different formulations?
- Why the percentage of crystalline ITR in F1 was so lower than that in F2 and F3?
- How are the results of DSC aligned with SEM micrographs?
8) In section 3.3, how the authors calculated MMAD for AmB and ITR separately while both are being loaded in the same AmB-ITR MPs. Also, how FPF < 5 μm (%) for both AmB and ITR exceeded 100% for all tested formulations
9) Line 388, “Since none of the investigated variables demonstrated a significant effect on antifungal efficacy”. From where the authors draws such conclusion.
Line 394, “EDX evidenced that the free crystals corresponded to unencapsulated ITR, due to the presence of chlorine”. Where EDX data.
Comments on the Quality of English LanguageMinor editing of English language required
Author Response
In this study, the authors tailored microparticles containing a high dose high dose of amphotericin B and itraconazole for inhalation in an attempt to manage severe fungal pulmonary infections. The idea of the work is interesting. However, the manuscript could not be accepted in the present form for the following points
- An HPLC chromatogram for simultaneous determination of AmB and ITR should be provided.
We thank the reviewer for the comment. We have amended the method as it was not clear previously. Two different runs at each wavelength (406 nm for AmB and 260 n for ITR) were used. The retention time for each drug was also different, 8 min for AmB and 20 min for ITR. The methods were previously validated in: Fernandez-Garcia, R.; Walsh, D.; O'Connell, P.; Slowing, K.; Raposo, R.; Paloma Ballesteros, M.; Jimenez-Cebrian, A.; Chamorro-Sancho, M.J.; Bolas-Fernandez, F.; Healy, A.M.; et al. Can amphotericin B and itraconazole be co-delivered orally? Tailoring oral fixed-dose combination coated granules for systemic mycoses. Eur J Pharm Biopharm 2023, 183, 74-91, doi:10.1016/j.ejpb.2023.01.003. Now this paragraph read as follows:
“A BDS Hypersil C18, 5 µm (200 x 4.6 mm) column (Thermo Fisher Scientific Inc., Massachusetts, USA) was utilised for separation, and UV detection wavelength was set at 406 and 260 nm for AmB and ITR, respectively, while the injection volume was 100 µl. Each sample was quantified in two different runs setting the corresponding wavelength for each drug. The retention time for AmB was 8 min while for the ITR was 20 min”.
- The description of results in Section 3.1. need to be revised regarding the impact (positive/negative effects) of different formulation variables on microparticle characteristics since Pareto charts did not support their explanation.
We thank the reviewer for this observation, in the figure legend the numbers were included behind each parameter what generates confusion. Now the legend for figure 1 reads as follows which matches the explanation provide in the text:
“Figure 1. Pareto charts describing the influence of the seven factors in the Taguchi design: a) Yield; b) Particle size; c) Inhibition halo; d) Encapsulation efficiency; e) Aggregation rate. The orange bar represents a positive effect, while the blue bar represents a negative effect. Key: (1) Amount of AmB, (2) Amount of ITR (3) Amount of γ-CD, (4) Amount of leucine , (5) Inlet temperature, (6) Gas flow rate, (7) Solution feed rate. Bonferroni limit: effects that are above the Bonferroni limit are almost certainly important; t-value limit: effects that are above the t-value limit are possibly important.”
- On what basis the authors selected the amount of both drugs (AmB and ITR), the amount of leucine, and the gas flow rate. As formulation variables for further Box Behnken design given that Factor such as inlet temperature which had certain impact on yield was not included.
We thank the reviewer for the comment. Based on our experience, we decided to prioritize the particle size over the yield considering that for lung deposition this is a critical factor. At higher inlet temperature, smaller particle size as powder is better dried. For this reason, we decided to keep constant the highest level tested of 150 ºC. For other factors, such as the amount of γ-CD was adjusted based on the total amount of drug added to the formulation in a 1:1 ratio (w:w).
- In Box Behnen design, statistical analysis data (ANOVA) along with quadratic equations must be provided to better explain the impact of different formulation variables on product responses.
All this information has been added into Supplementary material. The following sentence has been added in the text: “Statistical analysis and mathematical equations are included in Supplementary material.”
- In section 3.2.1., the authors claimed that “Evidence of chlorine was found on the crystals indicating that ITR might be partially encapsulated”. How the authors confirm this claim.
ITR is the only molecule that has Cl- groups in its structure. If ITR would be fully encapsulated within the microparticles, Cl-would not have been detected by EDX. No Cl- is detected on the surface of the microparticles, but it is found on the elongated crystals visualized by SEM.
- Is irregular non-smooth surface microparticles are suitable for inhalation and lung deposition.
Yes, it has been described in literature that corrugated particles have a better flow and lung deposition compared to particles with a smooth surface. See references below.
Han, C.S.; Kang, J.H.; Park, E.H.; Lee, H.J.; Jeong, S.J.; Kim, D.W.; Park, C.W. Corrugated surface microparticles with chitosan and levofloxacin for improved aerodynamic performance. Asian J Pharm Sci 2023, 18, 100815, doi:10.1016/j.ajps.2023.100815.
Scherliess, R.; Bock, S.; Bungert, N.; Neustock, A.; Valentin, L. Particle engineering in dry powders for inhalation. Eur J Pharm Sci 2022, 172, 106158, doi:10.1016/j.ejps.2022.106158.
Chew, N.Y.; Chan, H.K. Use of solid corrugated particles to enhance powder aerosol performance. Pharm Res 2001, 18, 1570-1577, doi:10.1023/a:1013082531394.
- In section 3.2.2., what the authors means by “The percentage of crystalline ITR was 5.79, 61, and 80% for F1, F2 and F3, respectively, which align with the SEM micrograph results.”- How the authors calculated the percentage of crystalline ITR in different formulations?- Why the percentage of crystalline ITR in F1 was so lower than that in F2 and F3?- How are the results of DSC aligned with SEM micrographs?
In the method section 2.2.3.2., at the end of the paragraph was indicated: “The percentage of ITR crystalline was calculated taking into account the enthalpy of fusion of unprocessed ITR and the amount of ITR in the sample”. Considering the fraction of ITR included in each sample and the enthalpy of fusion of the pure raw crystalline ITR as 100%, the percentage of crystalline ITR in each formulation was calculated. The F1 formulation is the one that contains the lower percentage of ITR and hence, mostly all drug is incorporated within the microparticulate system. However, at larger percentages of ITR such as with F2 and F3, the amount of crystalline encapsulate ITR was increased. The larger the crystallinity, the larger the presence of crystals visualized in the SEM images. We know the crystals correspond to ITR and no AmB because the presence of chlorine.
- In section 3.3, how the authors calculated MMAD for AmB and ITR separately while both are being loaded in the same AmB-ITR MPs. Also, how FPF < 5 μm (%) for both AmB and ITR exceeded 100% for all tested formulations.
Both drugs are calculated independently. We agree with the reviewer that this analysis can be confusing. This is why we added the following paragraph:
In all the cases, the FPF is calculated for each drug independently. Based on the results obtained, AmB and ITR-loaded microparticles have a suitable lung deposition pattern in deeper regions of the respiratory tract. Even though, there is a significant unencapsulated ITR fraction corresponding to the crystals observed in SEM, still possesses a particle size small enough to reach the lung parenchyma but in the upper regions.
- Line 388, “Since none of the investigated variables demonstrated a significant effect on antifungal efficacy”. From where the authors draws such conclusion.
Results are drawn from the Box-Behnken analysis. Only four variables were further investigated, including the amount of both drugs (AmB and ITR), the amount of leucine, and the gas flow rate. However, no statistically significant differences were observed for this response as this mainly is affected by antifungal concentration in solution when doing the test.
- Line 394, “EDX evidenced that the free crystals corresponded to unencapsulated ITR, due to the presence of chlorine”. Where EDX data.
EDX was a measurement just performed during the SEM analysis. Data was recorded numerically during the analysis but image was not incorporated within the main data set. To avoid confusion to the readers, the following sentence has been added to the text:
EDX evidenced that the free crystals corresponded to unencapsulated ITR, due to the presence of chlorine (data not shown).
Reviewer 2 Report
Comments and Suggestions for Authors
There are some concerns to be addressed:
It is much better for readers that optimization of AmB-ITR MPs in the experimental part be divided into sections with more organized flow of data presentation.
Equation 2 for encapsulation efficiency, the authors mentioned that A was the amount of drug (AmB or ITR) recovered and B was the total amount of drug in the formulation. I need more clarification what the meaning by AmB or ITR
Pulmonary pharmacokinetics is needed to verify whether entrapment of AmB-ITR within microparticles could prolong the localized action of AmB-ITR in lungs and, thereby, enhance therapeutic efficacy.
The optimization of the composition of AmB-ITR MPs was carried out using DoE, could the author clarify more about dependent and independent factors and selected responses as seen in Table 2
The author mentioned that the engineered formulations exhibited a complementary in vitro deposition pattern, corresponding to an upper airway ITR distribution along with a lower airway AmB deposition.
Efficient delivery and sufficient drug retention within the site of action are critical determinants of the effectiveness of targeted drug therapy. Accordingly, a bio-distribution study should be carried out to investigate AmB exposure in the organs following intrapulmonary administration.
In vitro release of either free AmB-ITR or AmB-ITR from microparticles should be conducted.
Stability indicating parameters such as visual appearance, drug loading and entrapment efficiency (%) should be estimated at regular intervals for the optimized formulation.
Comments on the Quality of English Language
Ok
Author Response
There are some concerns to be addressed:
It is much better for readers that optimization of AmB-ITR MPs in the experimental part be divided into sections with more organized flow of data presentation.
We thank to the reviewer for the observation. We have structured the experimental part better indicating clearly the different sections.
Equation 2 for encapsulation efficiency, the authors mentioned that A was the amount of drug (AmB or ITR) recovered and B was the total amount of drug in the formulation. I need more clarification what the meaning by AmB or ITR
We thank the reviewer for the observation. We have modified this paragraph and now the manuscript reads as follows:
“Where A was the amount of drug recovered and B was the total amount of drug included initially in the formulation which was considered as 100%. The encapsulation efficiency was calculated for AmB and ITR independently”.
Pulmonary pharmacokinetics is needed to verify whether entrapment of AmB-ITR within microparticles could prolong the localized action of AmB-ITR in lungs and, thereby, enhance therapeutic efficacy.
We thank the reviewer for this interesting observation. We are planning to perform PK experiments in a second stage. We have already evaluated the efficacy of AmB microparticles after direct lung administration (de Pablo, 2023) and results were promising with longer retention times in the lung parenchyma than other commercially available formulation such as AmBisome.
de Pablo, E.; O'Connell, P.; Fernandez-Garcia, R.; Marchand, S.; Chauzy, A.; Tewes, F.; Dea-Ayuela, M.A.; Kumar, D.; Bolas, F.; Ballesteros, M.P.; et al. Targeting lung macrophages for fungal and parasitic pulmonary infections with innovative amphotericin B dry powder inhalers. Int J Pharm 2023, 635, 122788, doi:10.1016/j.ijpharm.2023.122788.
The optimization of the composition of AmB-ITR MPs was carried out using DoE, could the author clarify more about dependent and independent factors and selected responses as seen in Table 2.
Based on our experience, we decided to prioritize the particle size over the yield considering that for lung deposition this is a critical factor. At higher inlet temperature, smaller particle size as powder is better dried. For this reason, we decided to keep constant the highest temperature tested of 150 ºC. For other factors, such as the amount of γ-CD was adjusted based on the total amount of drug added to the formulation in a 1:1 ratio (w:w). The independent factors are the amount of each drug, the amount of leucine and the gas flow rate, while the dependent factors were the different responses evaluated in this study.
The author mentioned that the engineered formulations exhibited a complementary in vitro deposition pattern, corresponding to an upper airway ITR distribution along with a lower airway AmB deposition. Efficient delivery and sufficient drug retention within the site of action are critical determinants of the effectiveness of targeted drug therapy. Accordingly, a bio-distribution study should be carried out to investigate AmB exposure in the organs following intrapulmonary administration. In vitro release of either free AmB-ITR or AmB-ITR from microparticles should be conducted.
We agree with the reviewer that this is a critical point to determine specially for clinical trials. Based on the NGI deposition pattern is evident that the unencapsulated fraction of ITR does not deposit in the same manner that the encapsulated fraction which can reach deeper areas of the respiratory tract. In previous studies performed with AmB microparticulate lung formulations (E. de Pablo, 2023), we observed that the AmB remained in the epithelial lining fluid over 24 h which is enough time to elicit a pharmacological effect. Also, in a second stage of experiments we are planning to evaluate the uptake by macrophages. It can be envisaged that the larger ITR unencapsulated crystals will be removed faster by the pulmonary macrophages which can result in a detrimental concentration. However, the IC50 for both AmB and ITR is low against different Candida spp, around 1 µg/ml, suggesting the suitability of the drug content in the formulations tested.
de Pablo, E.; O'Connell, P.; Fernandez-Garcia, R.; Marchand, S.; Chauzy, A.; Tewes, F.; Dea-Ayuela, M.A.; Kumar, D.; Bolas, F.; Ballesteros, M.P.; et al. Targeting lung macrophages for fungal and parasitic pulmonary infections with innovative amphotericin B dry powder inhalers. Int J Pharm 2023, 635, 122788, doi:10.1016/j.ijpharm.2023.122788.
Stability indicating parameters such as visual appearance, drug loading and entrapment efficiency (%) should be estimated at regular intervals for the optimized formulation.
After spray drying of the formulations, they were kept in the fridge under desiccated conditions. Before performing the different assays, a visual inspection as well as the drug loading was quantified showing up to 6 months greater values than 95% with no visual change in the particle morphology. This data has not been included in the manuscript as a thorough experiment was not performed.
Reviewer 3 Report
Comments and Suggestions for Authors
The manuscript entitled "Co-delivery of a high dose of amphotericin B and itraconazole by means of a dry powder inhaler formulation for the treatment of severe fungal pulmonary infections" presents interesting results. Authors have objectively presented the data and discussed in detail. The manuscript is suitable after minor revisions.
1. Some technical issues in the presentation are there. e,g. subscript in H3PO4 throughout the manuscript.
2. The resolution of figure 1 is poor making it hard to read the contents. Please improve.
3. Figure 5a; X-axis content needs improvement.
4. The discussion section can be improved. More details of methods like SEM etc can be included.
Comments on the Quality of English LanguageMinor editing required
Author Response
The manuscript entitled "Co-delivery of a high dose of amphotericin B and itraconazole by means of a dry powder inhaler formulation for the treatment of severe fungal pulmonary infections" presents interesting results. Authors have objectively presented the data and discussed in detail. The manuscript is suitable after minor revisions.
- Some technical issues in the presentation are there. e,g. subscript in H3PO4 throughout the manuscript.
We thank the reviewer for this observation. This has now been amended in the text.
- The resolution of figure 1 is poor making it hard to read the contents. Please improve.
The resolution of Figure 1 has now been improved.
- Figure 5a; X-axis content needs improvement.
The axis content has been improved in the manuscript.
- The discussion section can be improved. More details of methods like SEM etc can be included.
The following paragraphs have now been included:
“In all the cases, the FPF is calculated for each drug independently. Based on the resutls obtained, AmB and ITR-loaded microparticles have a suitable lung deposition pattern in deeper regions of the respiratory tract. Even though, there is a significant unencapsulated ITR fraction corresponding to the crystals observed in SEM, still possesses a particle size small enough to reach the lung parenchyma but in the upper regions”.
“ITR is the only molecule that has Cl- groups in its structure. If ITR would be fully encapsulated within the microparticles, Cl- would not have been detected by EDX. No chlorine was detected on the surface of the microparticles, while it was found on the elongated crystals visualized by SEM”
Round 2
Reviewer 1 Report
Comments and Suggestions for Authors
Minor editing of English language required
Author Response
The description of results in Section 3.1. need to be revised regarding the impact (positive/negative effects) of different formulation variables on microparticle characteristics since Pareto charts dis not support their explanation.
The response of the authors is not satisfactory; where, for example, in Figure 1a, they mentioned that the amount of leucine and inlet temperature had a significant positive effect on yield, while the solution rate showed a negative effect. What about the remaining 4 factors out of tested 7 factors? Are they exerted synergistic or antagonistic effect.
We thank the reviewer for this observation. The reason why only some factors were highlighted was because they were the only ones who exhibited a significant effect. Nevertheless, the text has been amended, providing an explanation to each variable, as follows:
“The factors and responses from the Taguchi design are shown in Table 1. Pareto charts were constructed to understand the influence of each factor on each dependent variable (Fig. 1). At the tested conditions, the amount of leucine and inlet temperature had a significant positive effect on yield, while the solution feed rate showed a negative effect. However, although gas flow rate and the amount of ITR showed a positive effect on yield, while the amounts of γ-CD and AmB exhibited a negative effect, these factors were not significant (Fig. 1a). Both solution feed rate and the amount of ITR promoted a significant increase in particle size, while inlet temperature, the amount of leucine and gas flow rate had a significant impact in particle size, promoting a smaller particle size. The amount of AmB promoted an increase of particle size and the amount of γ-CD enhanced an opposite effect, nevertheless, the impact was not significant (Fig. 1b). Regarding in vitro antifungal efficacy, AmB-ITR MPs were more effective when higher gas flow rate, high amount of leucine and high solution feed rate were used, while the factors with a negative impact on antifungal efficacy were the amount of γ-CD and the gas flow rate (Fig. 1c); nevertheless, all formulations showed effectiveness against C. albicans (inhibition halo > 15 mm) (Table 1) [22]. The amount of AmB did not exhibit an increase in effectiveness, while the amount of ITR did, although these effects were not significant. Encapsulation efficiency was positively influenced by the amount of γ-CD and inlet temperature. The positive effect of γ-CD might be explained by the high entrapment capacity of the AmB within γ-CD. However, the amount of ITR and AmB had a negative impact on encapsulation efficiency, as higher amounts of the drug might be competing for entrapment within the γ-CD. The amount of leucine and the gas flow rate also exhibited a negative effect in encapsulation efficiency, although the effect was not significant in these cases (Fig. 1d). Finally, the amount of AmB and leucine triggered the dimeric aggregation state of AmB with a significant effect, meanwhile the dimerisation triggered by the amount of ITR and the solution feed rate was not significant. On the other hand, the inlet temperature, the amount of γ-CD and gas flow rate promoted a monomeric AmB, but with no significant effect (Fig. 1e).”
On what basis the authors selected of both drugs (AmB and ITR), the amount of leucine, and the gas flow rate. As formulation variables for further Box Behnken design given that Factor such as inlet temperature which had certain impact on yield was not included.
These data should be cited in the manuscript text rather than just putting explanation to the reviewer comment.
We thank the reviewer for the comment. The following statement has been added to the manuscript:
“These four factors were selected prioritising particle size over other variables, in which other factors might have shown a significant impact (e.g. inlet temperature showed a significant effect on yield), as particle size is critical for lung deposition. At higher inlet temperature, smaller particle size as powder is better dried. For this reason, we decided to keep constant the highest level tested of 150 °C.”
In section 3.2.1., the authors claimed that “Evidence of chlorine was found on the crystals indicating that ITR might be partially encapsulated”. How the authors confirm this claim.
Again, my question was how to identify Cl-groups in SEM images?
The SEM utilised has connected an energy dispersive X-Ray probe (EDX) that allows to perform an elemental analysis on the image visualised. ITR is the only molecule in the formulation with chlorine groups in its structure. As chlorine was detected by EDX outside the microparticles structure, that means ITR is not fully encapsulated. EDX spectra have been included in the supplementary material to support this information. The use of EDX is described in the methodology.
In section 3.2.2., what the authors means by “The percentage of crystalline ITR was 5.79, 61, and 80% for F1, F2 and F3, respectively, which align with the SEM micrograph results.” – How the authors calculated the percentage of crystalline ITR in different formulations? - Why the percentage of crystalline ITR in F1 was so lower than that in F2 and F3? - How are the results of DSC aligned with SEM micrographs?
Again, Why the percentage of crystalline ITR in F1 was so lower than that in F2 and F3? You mentioned that the percentage of crystalline ITR was 5.79, 61, and 80% for F1, F2 and F3, respectively. I think that variation in drug percentage from 10% ITR in F1 to 20% ITR in F2 will not cause such huge variation in percentage of crystalline ITR from 5.79 to 80%. Please clarify.
We have amended and discuss in depth this point. Now the text reads as follows:
The percentage of crystalline ITR was 5.79, 61, and 80% for F1, F2 and F3, respectively, which aligns with the SEM micrograph results. A higher density of crystals are observed when higher percentage of both antifungal drugs were used… β-CD has a smaller cavity than γ-CD which indicates that the latter is large enough to accommodate the ITR molecules; however, the amount of CD required to solubilise ITR is high. During the formulation process, AmB is first incorporated in the aqueous medium which can trigger the initial complexation between AmB and CD. ITR is added in a later step so only the free uncomplexed γ-CD is available for interaction. This results in a poorer solubilisation capacity and the failure to encapsulate all the ITR content, which is more evident when larger amounts of ITR are used as AmB is displacing ITR from its interaction with the CD, resulting in a significantly much higher percentage of crystalline ITR at higher total content of antimicrobials
In section 3.3, how the authors calculated MMAD for AmB and ITR separately while both are being loaded in the same AmB-ITR MPs. Also, how FPF < 5 µm (%) for both AmB and ITR exceeded 100% for all tested formulations.
Ok, I could understand that MMAD for AmB and ITR was calculated separately. How? Please provide raw calculation data.
The following sentence has been included in the manuscript: Data Analysis Software from Copley Scientific (Nottingham, United Kingdom) was used for MMAD analysis. The NGI data was analysed using a software provided by the company Copley Scientific in which based on the air flow used, the aerodynamic cut-off diameters (ACD) were: 8.06 µm, 4.46 µm, 2.82 µm, 1.66 µm, 0.94 µm, 0.55 µm, 0.34 µm and <0.34 µm from stage 1 to stage 8, respectively. Raw data from HPLC can be provide when requested.
Line 394, “EDX evidenced that the free crystals corresponded to unencapsulated ITR, due to the presence of chlorine”. Where EDX data.
The authors must include EDX data as supplementary data rather than saying “data not shown”.
EDX data have been included in the supplementary material.
Reviewer 2 Report
Comments and Suggestions for Authors
The authors addressed the essential raised points
Comments on the Quality of English Languageok
Author Response
The authors addressed the essential raised points.
We thank the reviewer for the feedback provided. Grammar has been carefully checked on avoid any mistakes or typos.
Round 3
Reviewer 1 Report
Comments and Suggestions for Authors
The authors had responded to all reviewer comments and the manuscript can be accepted in the current form.
Comments on the Quality of English LanguageMinor editing of English language required